# Dynamic Sparse Training: Find Efficient Sparse Network from scratch with Trainable Masked Layers

**Junjie LIU[1], Zhe XU[2], Runbin SHI[1], Ray C.C. CHEUNG[2] & Hayden K.H. SO[1]**
[1]Department of Electrical and Electronic Engineering, The University of Hong Kong
[2]Department of Electrical Engineering, City University of Hong Kong
`{jjliu,rbshi,hso}@eee.hku.hk`
`zhexu22-c@my.cityu.edu.hk, r.cheung@cityu.edu.hk`

## Abstract

We present a novel network pruning algorithm called Dynamic Sparse Training that can jointly find the optimal network parameters and sparse network structure in a unified optimization process with trainable pruning thresholds. These thresholds can have fine-grained layer-wise adjustments dynamically via back-propagation. We demonstrate that our dynamic sparse training algorithm can easily train very sparse neural network models with little performance loss using the same number of training epochs as dense models. Dynamic Sparse Training achieves state of the art performance compared with other sparse training algorithms on various network architectures. Additionally, we have several surprising observations that provide strong evidence to the effectiveness and efficiency of our algorithm. These observations reveal the underlying problems of traditional three-stage pruning algorithms and present the potential guidance provided by our algorithm to the design of more compact network architectures.

## 1 Introduction

Despite the impressive success that deep neural networks have achieved in a wide range of challenging tasks, the inference in deep neural networks is highly memory-intensive and computation-intensive due to the over-parameterization of deep neural networks. Network pruning (LeCun et al. (1990); Han et al. (2015); Molchanov et al. (2017)) has been recognized as an effective approach to improving the inference efficiency in resource-limited scenarios.

Traditional pruning methods consist of dense network training followed with pruning and fine-tuning iterations. To avoid the expensive pruning and fine-tuning iterations, many sparse training methods (Mocanu et al., 2018; Bellec et al., 2017; Mostafa & Wang, 2019; Dettmers & Zettlemoyer, 2019) have been proposed, where the network pruning is conducted during the training process. However, all these methods suffer from following three problems:

**Coarse-grained predefined pruning schedule**. Most of the existing pruning methods use a predefined pruning schedule with many additional hyperparameters like pruning $a\%$ parameter each time and then fine-tuning for $b$ epochs with totally $c$ pruning steps. It is non-trivial to determine these hyperparameters for network architectures with various degrees of complexity. Therefore, usually a fixed pruning schedule is adopted for all the network architectures, which means that a very simple network architecture like LeNet-300-100 will have the same pruning schedule as a far more complex network like ResNet-152. Besides, almost all the existing pruning methods conduct epoch-wise pruning, which means that the pruning is conducted between two epochs and no pruning operation happens inside each epoch.

**Failure to properly recover the pruned weights**. Almost all the existing pruning methods conduct "hard" pruning that prunes weights by directly setting their values to 0. Many works (Guo et al., 2016; Mocanu et al., 2018; He et al., 2018) have argued that the importance of network weights are not fixed and will change dynamically during the pruning and training process. Previously unimportant weights may tend to be important. So the ability to recover the pruned weights is of

high significance. However, directly setting the pruned weights to 0 results in the loss of historical parameter importance, which makes it difficult to determine: 1) whether and when each pruned weight should be recovered, 2) what values should be assigned to the recovered weights. Therefore, existing methods that claim to be able to recover the pruned weights simply choose a predefined portion of pruned weights to recover and these recover weights are randomly initialized or initialized to the same value.

**Failure to properly determine layer-wise pruning rates**. Modern neural network architectures usually contain dozens of layers with a various number of parameters. Therefore, the degree of parameter redundancy is very different among the layers. For simplicity, some methods prune the same percentage of parameters at each layer, which is not optimal. To obtain dynamic layer-wise pruning rates, a single global pruning threshold or layer-wise greedy algorithms are applied. Using a single global pruning threshold is exceedingly difficult to assess the local parameter importance of the individual layer since each layer has a significantly different amount of parameter and contribution to the model performance. This makes pruning algorithms based on a single global threshold inconsistent and non-robust. The problem of layer-by-layer greedy pruning methods is that the unimportant neurons in an early layer may have a significant influence on the responses in later layers, which may result in propagation and amplification of the reconstruction error (Yu et al., 2018).

We propose a novel end-to-end sparse training algorithm that properly solves the above problems. With only one additional hyperparameter used to set the final model sparsity, our method can achieve dynamic fine-grained pruning and recovery during the whole training process. Meanwhile, the layer-wise pruning rates will be adjusted automatically with respect to the change of parameter importance during the training and pruning process. Our method achieves state-of-the-art performance compared with other sparse training algorithms. The proposed algorithm has following promising properties:

• **Step-wise pruning and recovery.** A training epoch usually will have tens of thousands of training steps, which is the feed-forward and back-propagation pass for a single mini-batch. Instead of pruning between two training epochs with a predefined pruning schedule, our method prunes and recovers the network parameter at each training step, which is far more fine-grained than existing methods.

• **Neuron-wise or filter-wise trainable thresholds.** All the existing methods adopt a single pruning threshold for each layer or the whole architecture. Our method defines a threshold vector for each layer. Therefore, our method adopts neuron-wise pruning thresholds for fully connected and recurrent layer and filter-wise pruning thresholds for convolutional layer. Additionally, all these pruning thresholds are trainable and will be updated automatically via back-propagation.

• **Dynamic pruning schedule.** The training process of deep neural network consists of many hyperparameters. The learning rate is perhaps the most important hyperparameter. Usually, the learning rate will decay during the training process. Our method can automatically adjust the layer-wise pruning rates under different learning rates to get the optimal sparse network structure.

• **Consistent sparse pattern.** Our algorithm can get a consistent layer-wise sparse pattern under different model sparsities, which indicates that our method can automatically determine the optimal layer-wise pruning rates given the target model sparsity.

## 2 RELATED WORK

**Traditional Pruning Methods**: LeCun et al. (1990) presented the early work about network pruning using second-order derivatives as the pruning criterion. The effective and popular training, pruning and fine-tuning pipeline was proposed by Han et al. (2015), which used the parameter magnitude as the pruning criterion. Narang et al. (2017) extended this pipeline to prune the recurrent neural networks with a complicated pruning strategy. Molchanov et al. (2016) introduced first-order Taylor term as the pruning criterion and conduct global pruning. Li et al. (2016) used $\ell_1$ regularization to force the unimportant parameters to zero.

**Sparse Neural Network Training**: Recently, some works attempt to find the sparse network directly during the training process without the pruning and fine-tuning stage. Inspired by the growth and extinction of neural cells in biological neural networks, Mocanu et al. (2018) proposed a prune-regrowth procedure called Sparse Evolutionary Training (SET) that allows the pruned neurons and

connections to revive randomly. However, the sparsity level needs to be set manually and the random recovery of network connections may provoke unexpected effects on the network. DEEP-R proposed by Bellec et al. (2017) used Bayesian sampling to decide the pruning and regrowth configuration, which is computationally expensive. Dynamic Sparse Reparameterization (Mostafa & Wang, 2019) used dynamic parameter reallocation to find the sparse structure. However, the pruning threshold can only get halved if the percentage of parameter pruned is too high or get doubled if that percentage is too low for a certain layer. This coarse-grained adjustment of the pruning threshold significantly limits the ability of Dynamic Sparse Reparameterization. Additionally, a predefined pruning ratio and fractional tolerance are required. Dynamic Network Surgery (Guo et al., 2016) proposed pruning and splicing procedure that can prune or recover network connections according to the parameter magnitude but it requires manually determined thresholds that are fixed during the sparse learning process. These layer-wise thresholds are extremely hard to manually set. Meanwhile, Fixing the thresholds makes it hard to adapt to the rapid change of parameter importance. Dettmers & Zettlemoyer (2019) proposed sparse momentum that used the exponentially smoothed gradients as the criterion for pruning and regrowth. A fixed percentage of parameters are pruned at each pruning step. The pruning ratio and momentum scaling rate need to be searched from a relatively high parameter space.

## 3 DYNAMIC SPARSE TRAINING

### 3.1 NOTATION

Deep neural network consists of a set of parameters $\{\boldsymbol{W}_i : 1 \leq i \leq C\}$, where $\boldsymbol{W}_i$ denotes the parameter matrix at layer $i$ and $C$ denotes the number of layers in this network. For each fully connected layer and recurrent layer, the corresponding parameter is $\boldsymbol{W}_i \in \mathbb{R}^{c_o \times c_i}$, where $c_o$ is the output dimension and $c_i$ is the input dimension. For each convolutional layer, there exists a convolution kernel $\mathsf{K}_i \in \mathbb{R}^{c_o \times c_i \times w \times h}$, where $c_o$ is the number of output channels, $c_i$ is the number of input channels, $w$ and $h$ are the kernel sizes. Each filter in a convolution kernel $\mathsf{K}_i$ can be flattened to a vector. Therefore, a corresponding parameter matrix $\boldsymbol{W}_i \in \mathbb{R}^{c_o \times z}$ can be derived from each convolution kernel $\mathsf{K}_i \in \mathbb{R}^{c_o \times c_i \times w \times h}$, where $z = c_i \times w \times h$. Actually, the pruning process is equivalent to finding a binary parameter mask $\boldsymbol{M}_i$ for each parameter matrix $\boldsymbol{W}_i$. Thus, a set of binary parameter masks $\{\boldsymbol{M}_i : 1 \leq i \leq C\}$ will be found by network pruning. Each element for these parameter masks $\boldsymbol{M}_i$ is either 1 or 0.

### 3.2 THRESHOLD VECTOR AND DYNAMIC PARAMETER MASK

Pruning can be regarded as applying a binary mask $\boldsymbol{M}$ to each parameter $\boldsymbol{W}$. This binary parameter mask $\boldsymbol{M}$ preserves the information about the sparse structure. Given the parameter set $\{\boldsymbol{W}_1, \boldsymbol{W}_2, \cdots, \boldsymbol{W}_C\}$, our method will dynamically find the corresponding parameter masks $\{\boldsymbol{M}_1, \boldsymbol{M}_2, \cdots, \boldsymbol{M}_C\}$. To achieve this, for each parameter matrix $\boldsymbol{W} \in \mathbb{R}^{c_o \times c_i}$, a trainable pruning threshold vector $\boldsymbol{t} \in \mathbb{R}^{c_o}$ is defined. Then we utilize a unit step function $S(x)$ as shown in Figure 2(a) to get the masks according to the magnitude of parameters and corresponding thresholds as present below.

$$\boldsymbol{Q}_{ij} = F(\boldsymbol{W}_{ij}, t_i) = |\boldsymbol{W}_{ij}| - \boldsymbol{t}_i \quad 1 \leq i \leq c_o, 1 \leq j \leq c_i \tag{1}$$

$$\boldsymbol{M}_{ij} = S(\boldsymbol{Q}_{ij}) \quad 1 \leq i \leq c_o, 1 \leq j \leq c_i \tag{2}$$

With the dynamic parameter mask $\boldsymbol{M}$, the corresponding element in mask $\boldsymbol{M}_{ij}$ will be set to 0 if $\boldsymbol{W}_{ij}$ needs to be pruned. This means that the weight $\boldsymbol{W}_{ij}$ is masked out by the 0 at $\boldsymbol{M}_{ij}$ to get a sparse parameter $\boldsymbol{W} \odot \boldsymbol{M}$. The value of underlying weight $\boldsymbol{W}_{ij}$ will not change, which preserves the historical information about the parameter importance.

For a fully connected layer or recurrent layer with parameter $\boldsymbol{W} \in \mathbb{R}^{c_o \times c_i}$ and threshold vector $\boldsymbol{t} \in \mathbb{R}^{c_o}$, each weight $\boldsymbol{W}_{ij}$ will have a neuron-wise threshold $\boldsymbol{t}_i$, where $\boldsymbol{W}_{ij}$ is the $j$th weight associated with the $i$th output neuron. Similarly, the thresholds are filter-wise for convolutional layer. Besides, a threshold matrix or a single scalar threshold can also be chosen. More details are present in Appendix A.2.

## 3.3 TRAINABLE MASKED LAYERS

With the threshold vector and dynamic parameter mask, the trainable masked fully connected, convolutional and recurrent layer are introduced as shown in Figure 1, where $X$ is the input of current layer and $Y$ is the output. For fully connected and recurrent layers, instead of the dense parameter $W$, the sparse product $W \odot M$ will be used in the batched matrix multiplication, where $\odot$ denote Hadamard product operator. For convolutional layers, each convolution kernel $\mathbf{K} \in \mathbb{R}^{c_o \times c_i \times w \times h}$ can be flatten to get $W \in \mathbb{R}^{c_o \times z}$. Therefore, the sparse kernel can be obtained by a similar process as fully connected layers. This sparse kernel will be used for subsequent convolution operation.

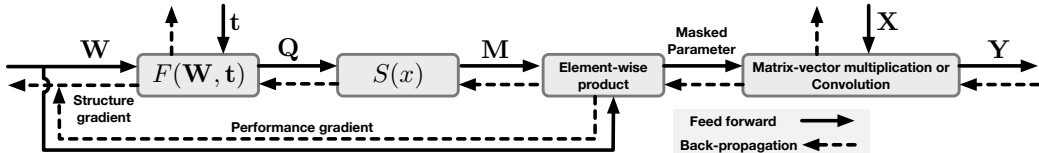

Figure 1: Detailed structure of trainable masked layer

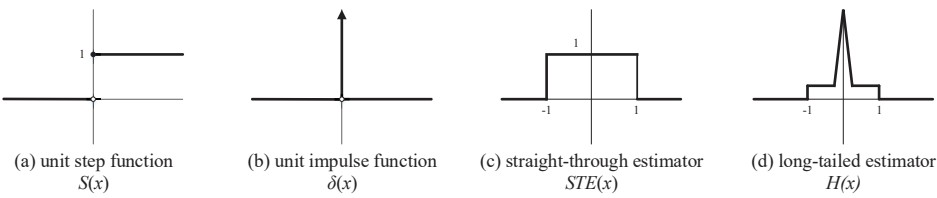

(a) unit step function
$S(x)$

(b) unit impulse function
$\delta(x)$

(c) straight-through estimator
$STE(x)$

(d) long-tailed estimator
$H(x)$

Figure 2: The unit step function $S(x)$ and its derivative approximations.

In order to make the elements in threshold vector $t$ trainable via back-propagation, the derivative of the binary step function $S(x)$ is required. However, its original derivative is an impulse function whose value is zero almost everywhere and infinite at zero as shown in Figure 2(b). Thus the original derivative of the binary step function $S(x)$ cannot be applied in back-propagation and parameter updating directly. Some previous works (Hubara et al. (2016); Rastegari et al. (2016); Zhou et al. (2016)) demonstrated that by providing a derivative estimation, it is possible to train networks containing such binarization function. A clip function called straight-through estimator (STE) (Bengio et al., 2013) was employed in these works and is illustrated in Figure 2(c).

Furthermore, Xu & Cheung (2019) discussed the derivative estimation in a balance of tight approximation and smooth back-propagation. We adopt this long-tailed higher-order estimator $H(x)$ in our method. As shown in Figure 2(d), it has a wide active range between $[-1, 1]$ with a non-zero gradient to avoid gradient vanishing during training. On the other hand, the gradient value near zero is a piecewise polynomial function giving tighter approximation than STE. The estimator is represented as

$$\frac{d}{dx}S(x) \approx H(x) = \begin{cases} 2 - 4|x|, & -0.4 \leq x \leq 0.4 \\ 0.4, & 0.4 < |x| \leq 1 \\ 0, & otherwise \end{cases} \tag{3}$$

With this derivative estimator, the elements in the vector threshold $t$ can be trained via back-propagation. Meanwhile, in trainable masked layers, the network parameter $W$ can receive two branches of gradient, namely the performance gradient for better model performance and the structure gradient for better sparse structure, which helps to properly update the network parameter under sparse network connectivity. The structure gradient enables the pruned (masked) weights to be updated via back-propagation. The details about the feed-forward and back-propagation of the trainable masked layer are present in Appendix A.3.

Therefore, the pruned (masked) weights, the unpruned (unmasked) weights and the elements in the threshold vector can all be updated via back-propagation at each training step. The proposed method will conduct fine-grained step-wise pruning and recovery automatically.

### 3.4 SPARSE REGULARIZATION TERM

Now that the thresholds of each layer are trainable, a higher percentage of pruned parameter is desired. To get the parameter masks $M$ with high sparsity, higher pruning thresholds are needed. To achieve this, we add a sparse regularization term $L_s$ to the training loss that penalizes the low threshold value. For a trainable masked layer with threshold $t \in \mathbb{R}^{c_o}$, the corresponding regularization term is $R = \sum_{i=1}^{c_o} \exp(-t_i)$. Thus, the sparse regularization term $L_s$ for a deep neural network with $C$ trainable masked layers is:

$$L_s = \sum_{i=1}^{C} R_i \tag{4}$$

We use $\exp(-x)$ as the regularization function since it is asymptotical to zero as $x$ increases. Consequently, it penalizes low thresholds without encouraging them to become extremely large.

### 3.5 DYNAMIC SPARSE TRAINING

The traditional fully connected, convolutional and recurrent layers can be replaced with the corresponding trainable masked layers in deep neural networks. Then we can train a sparse neural network directly with back-propagation algorithm given the training dataset $\mathcal{D} = \{(x_1, y_1), (x_2, y_2), \cdots, (x_N, y_N)\}$, the network parameter $W$ and the layer-wise threshold $t$ as follows:

$$\mathcal{J}(W, t) = \frac{1}{N}(\sum_{i=1}^{N} \mathcal{L}((x_i, y_i); W, t)) + \alpha L_s \tag{5}$$

$$W^*, t^* = \underset{W, t}{\arg\min} \, \mathcal{J}(W, t) \tag{6}$$

where $\mathcal{L}(\cdot)$ is the loss function, e.g. cross-entropy loss for classification and $\alpha$ is the scaling coefficient for the sparse regularization term, which can control the percentage of parameter remaining. The sparse regularization term $L_s$ tends to increase the threshold $t$ for each layer thus getting higher model sparsity. However, higher sparsity tends to increase the loss function, which reversely tends to reduce the threshold and level of sparsity. Consequently, the training process of the thresholds can be regarded as a contest between the sparse regularization term and the loss function in the sense of game theory. Therefore, our method can dynamically find the sparse structure that properly balances the model sparsity and performance.

## 4 EXPERIMENTS

The proposed method is evaluated on MNIST, CIFAR-10 and ImageNet with various modern network architectures including fully connected, convolutional and recurrent neural networks. To quantify the pruning performance, the layer remaining ratio is defined to be $k_l = n/m$, where $n$ is the number of elements equal to $1$ in the mask $M$ and $m$ is the total number of elements in $M$. The model remaining ratio $k_m$ is the overall ratio of the non-zero elements in the parameter masks for all trainable masked layers. The model remaining percentage is defined as $k_m \times 100\%$. For all trainable masked layers, the trainable thresholds are initialized to zero since it is assumed that originally the network is dense. The detailed experimental setup is present in Appendix A.1.

### 4.1 PRUNING PERFORMANCE ON VARIOUS DATASETS

**MNIST.** Table 1 presents the pruning results of proposed method for Lenet-300-100 (LeCun et al., 1998), Lenet-5-Caffe and LSTM (Hochreiter & Schmidhuber, 1997). Both LSTM models have two LSTM layers with hidden size 128 for LSTM-a and 256 for LSTM-b. Our method can prune almost 98% parameter with little loss of performance on Lenet-300-100 and Lenet-5-Caffe. For the LSTM models adapted for the sequential MNIST classification, our method can find sparse models with better performance than dense baseline with over 99% parameter pruned.

**CIFAR-10.** The pruning performance of our method on CIFAR-10 is tested on VGG (Simonyan & Zisserman, 2014) and WideResNet (Zagoruyko & Komodakis, 2016). We compare our method

| Architecture | Dense Baseline (%) | Model Remaining Percentage (%) | Sparse Accuracy (%) | Difference |
|---|---|---|---|---|
| Lenet-300-100 | $98.16 \pm 0.06$ | $2.48 \pm 0.21$ | $97.69\pm0.14$ | -0.47 |
| Lenet-5-Caffe | $99.18 \pm 0.05$ | $1.64 \pm 0.13$ | $99.11\pm0.07$ | -0.07 |
| LSTM-a | $98.64 \pm 0.12$ | $1.93 \pm 0.03$ | $98.70\pm0.06$ | **+0.06** |
| LSTM-b | $98.87 \pm 0.07$ | $0.98 \pm 0.04$ | $98.89\pm0.11$ | **+0.02** |

Table 1: The pruning results on MNIST for various architectures

with other sparse learning algorithms on CIFAR-10 as presented in Table 2. The state-of-the-art algorithms, DSR (Mostafa & Wang, 2019) and Sparse momentum (Dettmers & Zettlemoyer, 2019), are selected for comparison. Dynamic Sparse Training (DST) outperforms them in almost all the settings as present in Table 2.

| Architecture | Method | Dense baseline | Model Remaining Percentage (%) | Sparse Accuracy | Difference |
|---|---|---|---|---|---|
| VGG-16 | Sparse Momentum | $93.51 \pm 0.05$ | 10 | $93.36 \pm 0.04$ | -0.15 |
| | | | 5 | $93.00 \pm 0.07$ | -0.51 |
| | DST (Ours) | $93.75 \pm 0.21$ | $8.82 \pm 0.34$ | **$93.93 \pm 0.05$** | **+0.18** |
| | | | $3.76 \pm 0.53$ | **$93.02 \pm 0.37$** | -0.73 |
| WideResNet-16-8 | Sparse Momentum | $95.43 \pm 0.02$ | 10 | $94.87 \pm 0.04$ | -0.56 |
| | | | 5 | $94.38 \pm 0.05$ | -1.05 |
| | DSR | $95.21 \pm 0.05$ | 10 | $94.93 \pm 0.04$ | -0.28 |
| | | | 5 | $94.68 \pm 0.05$ | -0.53 |
| | DST (Ours) | $95.18 \pm 0.06$ | $9.86 \pm 0.22$ | **$95.05 \pm 0.08$** | **-0.13** |
| | | | $4.64 \pm 0.15$ | **$94.73 \pm 0.11$** | **-0.45** |

Table 2: Comparison with other sparse training methods on CIFAR-10.

**ImageNet.** For ResNet-50 on ImageNet (ILSVRC2012), we present the performance and comparison with other methods in Table 3. Dynamic Sparse Training (DST) achieves better top-1 and top-5 accuracy with slightly higher model sparsity. The number of parameters for the dense ResNet-50 model is 25.56 M.

| Method | Dense baseline Top-1 / Top-5 (%) | Model Remaining Percentage (%) | Sparse accuracy Top-1 / Top-5 (%) | Difference | Remaining number of parameters |
|---|---|---|---|---|---|
| Sparse Momentum | 74.90 / 92.40 | 20 | 73.80 / 91.80 | -1.10 / -0.60 | 5.12M |
| | | 10 | 72.30 / 91.00 | -2.60 / -1.40 | 2.56M |
| DSR | 74.90 / 92.40 | 20 | 73.30 / 92.40 | -1.60 / +0.00 | 5.14M |
| | | 10 | 71.60 / 90.50 | -3.30 / -1.90 | 2.56M |
| DST (Ours) | 74.95 / 92.60 | 19.24 | **74.02 / 92.49** | **-0.73 / -0.11** | 5.08M |
| | | 9.87 | **72.78 / 91.53** | **-2.17 / -1.07** | 2.49M |

Table 3: Comparison with other sparse training methods for ResNet-50 on ImageNet.

## 4.2 Pruning performance on various remaining ratio

By varying the scaling coefficient $\alpha$ for sparse regularization term, we can control the model remaining ratios of sparse models generated by dynamic sparse training. The relationships between $\alpha$, model remaining ratio and sparse model accuracy of VGG, WideResNet-16-8 and WideResNet-28-8 on CIFAR-10 are presented in Figure 3. As demonstrated, the model remaining ratio keeps decreasing with increasing $\alpha$. With a moderate $\alpha$ value, it is easy to obtain a sparse model with comparable or even higher accuracy than the dense counterpart. On the other side, if the $\alpha$ value is too large that makes the model remaining percentage less than 5%, there will be a noticeable accuracy drop. As demonstrated in Figure 3, the choice of $\alpha$ ranges from $10^{-9}$ to $10^{-4}$. Depending on the application scenarios, we can either get models with similar or better performance as dense counterparts by a relatively small $\alpha$ or get a highly sparse model with little performance loss by a larger $\alpha$.

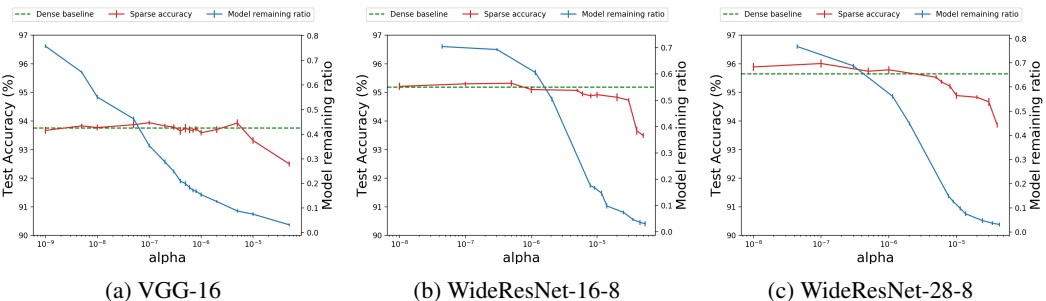

(a) VGG-16      (b) WideResNet-16-8      (c) WideResNet-28-8

Figure 3: Test accuracy of sparse model on CIFAR-10 and model remaining ratio for different $\alpha$

# 5 DISCUSSION

## 5.1 FINE-GRAINED STEP-WISE PRUNING

Figure 4(a) demonstrates the change of layer remaining ratios for Lenet-300-100 trained with DST at each training step during the first training epoch. And Figure 4(b) presents the change of layer remaining ratios during the whole training process (20 epochs). As present in these two figures, instead of decreasing by manually set fixed step size as in other pruning methods, our method makes the layer remaining ratios change smoothly and continuously at each training step. Meanwhile, as shown in Figure 4(a), the layer remaining ratios fluctuate dynamically within the first 100 training steps, which indicates that DST can achieve step-wise fine-grained pruning and recovery.

Meanwhile, for multilayer neural networks, the parameters in different layers will have different relative importance. For example, Lenet-300-100 has three fully connected layers. Changing the parameter of the last layer (layer 3) can directly affect the model output. Thus, the parameter of layer 3 should be pruned more carefully. The input layer (layer 1) has the largest amount of parameters and takes the raw image as input. Since the images in MNIST dataset consist of many unchanged pixels (the background) that have no contribution to the classification process, it is expected that the parameters that take these invariant background pixels as input can be pruned safely. Therefore, the remaining ratio should be the highest for layer 3 and the lowest for layer 1 if a Lenet-300-100 model is pruned. To check the pruning effect of our method on these three layers, Lenet-300-100 model is sparsely trained by the default hyperparameters setting present in Appendix A.1. The pruning trends of these three layers during dynamic sparse training are present in Figure 4(b). During the whole sparse training process, the remaining ratios of layer 1 and layer 2 keep decreasing and the remaining ratio of layer 3 maintains to be 1 after the fluctuation during the first epoch. The remaining ratio of layer 1 is the lowest and decrease to less than 10% quickly, which is consistent with the expectation. In the meantime, the test accuracy of the sparse model is almost the same as the test accuracy on the dense model in the whole training process. This indicates that our method can properly balance the model remaining ratio and the model performance by continuous fine-grained pruning throughout the entire sparse training procedure. A similar training tendency can be observed in other network architectures. The detailed results for other architectures are present in Appendix A.4.

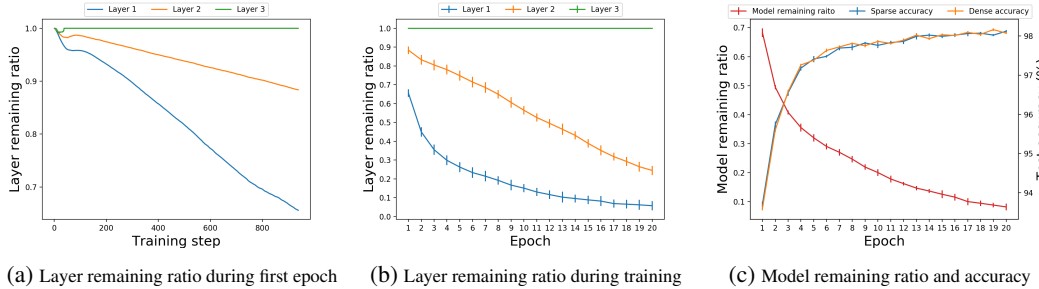

(a) Layer remaining ratio during first epoch    (b) Layer remaining ratio during training    (c) Model remaining ratio and accuracy

Figure 4: Change of layer remaining ratio, model remaining ratio and test accuracy for Lenet-300-100 with $\alpha = 0.0005$.

## 5.2 DYNAMIC PRUNING SCHEDULE

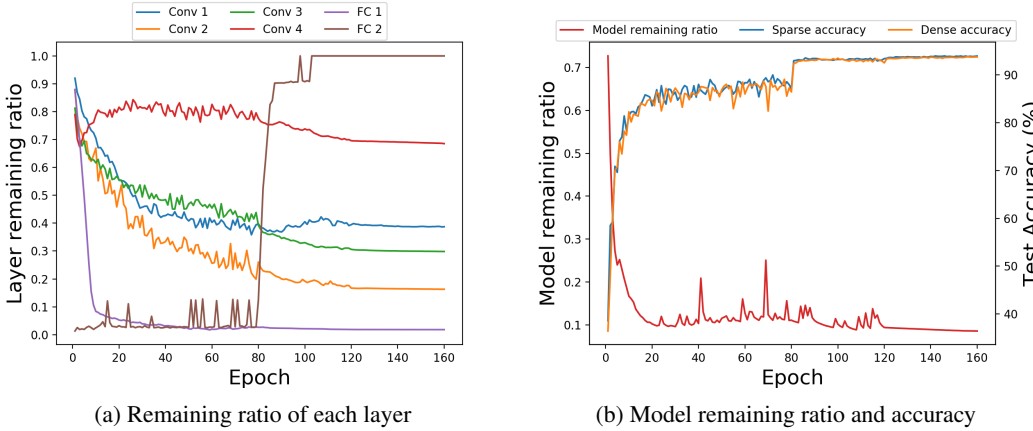

(a) Remaining ratio of each layer

(b) Model remaining ratio and accuracy

Figure 5: VGG-16 trained with initial learning rate 0.1 and $\alpha = 5 \times 10^{-6}$.

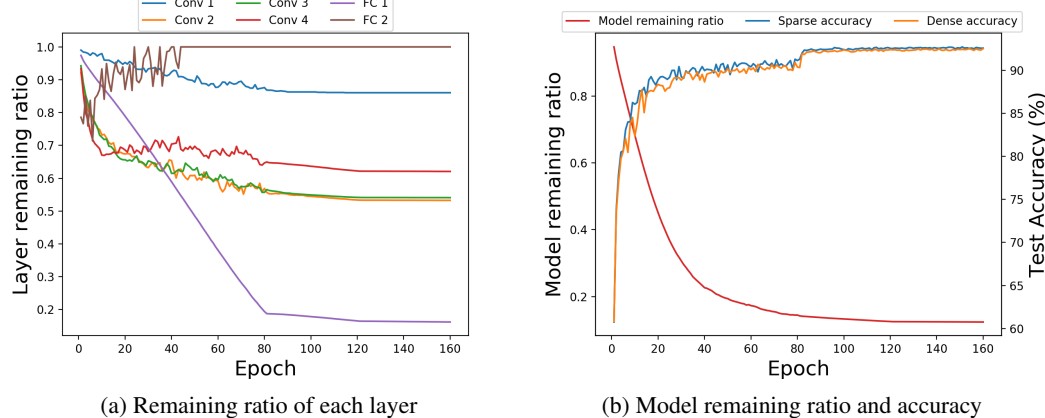

(a) Remaining ratio of each layer

(b) Model remaining ratio and accuracy

Figure 6: VGG-16 trained with initial learning rate 0.01 and $\alpha = 5 \times 10^{-6}$.

**Dynamic adjustment regarding learning rate decay.** To achieve better performance, it is a common practice to decay the learning rate several times during the training process of deep neural networks. Usually, the test accuracy will decrease immediately just after the learning rate decay and then tend toward flatness. A similar phenomenon is observed for VGG-16 on CIFAR-10 trained by DST as present in Figure 5(b), where the learning rate decay from 0.1 to 0.01 at 80 epoch.

Like the layer 3 of Lenet-300-100, the second fully connected layer (FC 2) of VGG-16 is the output layer, hence its remaining ratio is expected to be relatively high. But a surprising observation is that the remaining ratio of FC 2 is quite low at around 0.1 for the first 80 epochs and increases almost immediately just after the 80 epoch as present in Figure 5(a). We suppose that this is caused by the decay of the learning rate from 0.1 to 0.01 at 80 epoch. Before the learning rate decay, the layers preceding the output layer fail to extract enough useful features for classification due to the relatively coarse-grained parameter adjustment incurred by high learning rate. This means that the corresponding parameters that take those useless features as input can be pruned, which leads to the low remaining ratio of the output layer. The decayed learning rate enables a fine-grained parameter update that makes the neural network model converge to the good local minima quickly (Kawaguchi, 2016), where most of the features extracted by the preceding layers turn to be helpful for the classification. This makes previously unimportant network connections in the output layer become important thus the remaining ratio of this layer gets an abrupt increase.

There are two facts that support our assumptions for this phenomenon. The first fact is the sudden increase of the test accuracy from around 85% to over 92% just after the learning rate decay.

Secondly, the remaining ratios of the preceding convolutional layers are almost unchanged after the remaining ratio of the output layer increases up to 1, which means that the remaining parameters in these layers are necessary to find the critical features. The abrupt change of the remaining ratio of the output layer indicates that our method can dynamically adjust the pruning schedule regarding the change of hyperparameters during the training process.

**Dynamic adjustment regarding different initial learning rate.** To further investigate the effect of the learning rate, the VGG-16 model is trained by a different initial learning rate (0.01) with other hyperparameters unchanged. The corresponding training curves are presented in Figure 6. As shown in Figure 6(a), the remaining ratio of FC 2 increases up to 1 after around 40 epochs when the test accuracy reaches over 90%. This means that with the proper update of network parameters due to a smaller initial learning rate, the layers preceding the output layer (FC 2) tend to extract useful features before the learning rate decay. It can be observed in Figure 6(b) that the test accuracy only increases about 2% from 90% to 92% roughly after the learning rate decay at the 80 epoch. Meanwhile, one can see that our method adopts a different pruning schedule regarding different initial choices of training hyperparameters.

**Model performance under dynamic schedule.** Figure 5(b) and Figure 6(b) show the model remaining ratio, test accuracy of sparse model (sparse accuracy) and test accuracy of dense counterparts (dense accuracy) after each epoch during the whole training process. For the sparse model trained with initial learning rate 0.1, the final model remaining percentage is 8.82%. The test accuracy of this sparse model is 93.93%, which is better than 93.75% of the dense counterparts. Similarly, for the sparse model trained with initial learning rate 0.01, the final model remaining percentage is 12.45%. The test accuracy of this sparse model is 92.74%, which is also better than 92.54% of the dense model.

Considering the model performance during the whole training process, when trained with initial learning rate 0.01, the sparse accuracy is consistently higher than dense accuracy during the whole training process as present in Figure 6(b). Meanwhile, one can see from Figure 5(b) that the running average of sparse accuracy is also higher than that of dense accuracy during the whole training process when the initial learning rate is 0.1.

## 5.3 INFORMATION REVEALED FROM CONSISTENT SPARSE PATTERN

Many works have tried to design more compact models with mimic performance to over-parameterized models (He et al., 2016; Howard et al., 2017; Zhang et al., 2018). Network architecture search has been viewed as the future paradigm of deep neural network design. However, it is extremely difficult to determine whether the designed layer consists of redundancy. Therefore, typical network architecture search methods rely on evolutionary algorithms (Liu et al., 2017) or reinforcement learning (Baker et al., 2016), which is extremely time-consuming and computationally expensive. Network pruning can actually be reviewed as a kind of architecture search process (Liu et al., 2018; Frankle & Carbin, 2018) thus the sparse structure revealed from network pruning may provide some guidance to the network architecture design. However, The layer-wise equal remaining ratio generated by the unified pruning strategy fails to indicate the different degrees of redundancy for each layer. And the global pruning algorithm is non-robust, which fails to offer consistent guidance.

Here we demonstrate another interesting observation called consistent sparse pattern during dynamic sparse training that provides useful information about the redundancy of individual layers as the guidance for compact network architecture design. For the same architecture trained by our method with various $\alpha$ values, the relative relationship of sparsity among different layers keeps consistent. The sparse patterns of VGG-16 on CIFAR-10 are present in Figure 7 with three different $\alpha$.

In all configurations, the last four convolutional layers (Conv 10-13) and the first fully connected layers (FC 1) are highly sparse. Meanwhile, some layers (Conv 3-7, FC 2) keep a high amount of parameters after pruning. This consistent sparse pattern indicates that these heavily pruned layers consist of high redundancy and a large number of parameters can be reduced in these layers to get more compact models. This phenomenon also exists in other network architectures, which is present in detail in the appendix A.5. Therefore, with this consistent sparse pattern, after designing a new network architecture, our method can be applied to get useful information about the layer-wise redundancy in this new architecture.

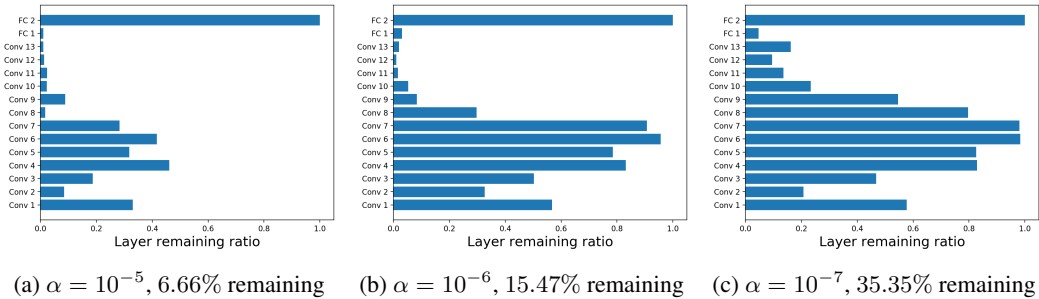

(a) $\alpha = 10^{-5}$, 6.66% remaining    (b) $\alpha = 10^{-6}$, 15.47% remaining    (c) $\alpha = 10^{-7}$, 35.35% remaining

Figure 7: The sparse pattern and percentage of parameter remaining for different choices of $\alpha$ on VGG-16

## 6 CONCLUSION

We propose Dynamic Sparse Training (DST) with trainable masked layers that enables direct training of sparse models with trainable pruning thresholds. DST can be easily applied to various types of neural network layers and architectures to get a highly sparse or better-performed model than dense counterparts. With the ability to reveal the consistent sparse pattern, our method can also provide useful guidance to the design of the more compact network.

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

# A APPENDIX

## A.1 EXPERIMENTAL SETUP

**MNIST**: LeNet-300-100 and LeNet-5-Caffe are trained using SGD with a momentum of 0.9 and the batch size of 64 with 20 training epochs. The learning rate is 0.01 without learning rate decay. Meanwhile, the default scaling factor $\alpha$ for sparse regularization term is 0.0005 for both network architectures. LSTM models are trained using Adam optimization scheme (Kingma & Ba, 2014) with default Adam hyperparameter setting for 20 epochs. The batch size is 100 and the default learning rate is 0.001. Meanwhile, the default scaling factor $\alpha$ for sparse regularization term is 0.001.

**CIFAR-10** Models on CIFAR-10 are trained using SGD with momentum 0.9 and batch size of 64 with 160 training epochs. The initial learning rate is 0.1 and decayed by 0.1 at 80 and 120 epoch. The default scaling factor $\alpha$ for sparse regularization term is $5 \times 10^{-6}$ for all tested architectures.

**ImageNet.** ResNet-50 models are trained using SGD with momentum 0.9 and batch size of 1024 with 90 training epochs. The default initial learning rate is 0.1 and decayed by 10 at 30, 60, 80 epochs.

For all trainable masked layers, the trainable thresholds are initialized to zero. Additionally, we find that an extremely high scaling coefficient $\alpha$ will make the sparse regularization term dominates the training loss thus making the mask $M$ to be all zero in a certain layer, which may impede the training process. To prevent this, the pruning threshold $t$ will be reset to zero if more than 99% elements in the mask are zero in a certain layer despite the LSTM models. This mechanism makes the training process smoother and enables a wider range choice of $\alpha$.

## A.2 ANALYSIS OF DIFFERENT THRESHOLD CHOICES

Beside a threshold vector, a threshold scalar $t$ or threshold matrix $T \in \mathbb{R}^{c_o \times c_i}$ can also be adopted for each trainable masked layer with parameter $W \in \mathbb{R}^{c_o \times c_i}$.

$$Q_{ij} = \begin{cases} |W_{ij}| - t, & \text{for threshold scalar} \\ |W_{ij}| - T_{ij}, & \text{for threshold matrix} \end{cases} \tag{7}$$

All three choices of trainable thresholds are tested on various architectures. Considering the effects on network pruning, the matrix threshold has almost the same model remaining ratio compared with the vector threshold. The scalar threshold is less robust than the vector and matrix threshold. In terms of the storage overhead, the matrix threshold almost doubles the amount of parameter during the training process in each architecture. Meanwhile, the vector threshold only adds less than 1% additional network parameter.

The extra trainable threshold will also bring additional computation. In both feed-forward and back-propagation phase, the additional computations are matrix-element-wise operations ($\mathcal{O}(n^2)$), which is apparently light-weighted compared to the original batched matrix multiplication ($\mathcal{O}(n^3)$). For practical deployment, only the masked parameter $W \odot M$ needs to be stored, thus no overhead will be introduced. Therefore, considering the balance between the overhead and the benefit incurred by these three choices, the vector threshold is chosen for trainable masked layers.

## A.3 MORE DETAILS ABOUT THE TRAINABLE MASKED LAYER

**Feed forward process.** Considering a single layer in deep neural network with input $x$ and dense parameter $W$. The normal layer will conduct matrix-vector multiplication as $Wx$ or convolution as $Conv(W, x)$. In trainable masked layers, since a sparse mask $M$ is obtained for each layer. The sparse parameter $W \odot M$ will be adopted in the corresponding matrix-vector multiplication or convolution as $(W \odot M)x$ and $Conv(W \odot M, x)$.

**Back-propagation process.** Referring to Figure 1, we denote $P = W \odot M$ and the gradient received by $P$ in back-propagation as $dP$. Considering the gradients from right to left:

- The performance gradient is $d\boldsymbol{P} \odot \boldsymbol{M}$
- The gradient received by $\boldsymbol{M}$ is $d\boldsymbol{P} \odot \boldsymbol{W}$
- The gradient received by $\boldsymbol{Q}$ is $d\boldsymbol{P} \odot \boldsymbol{W} \odot H(\boldsymbol{Q})$, where $H(\boldsymbol{Q})$ is the result of $H(x)$ applied to $\boldsymbol{Q}$ elementwisely.
- The structure gradient is $d\boldsymbol{P} \odot \boldsymbol{W} \odot H(\boldsymbol{Q}) \odot sgn(\boldsymbol{W})$, where $sgn(\boldsymbol{W})$ is the result of sign function applied to $W$ elementwisely.
- The gradient received by the vector threshold $\boldsymbol{t}$ is $d\boldsymbol{t} \in \mathbb{R}^{c_o}$. We denote $d\boldsymbol{T} = -d\boldsymbol{P} \odot \boldsymbol{W} \odot H(\boldsymbol{Q})$, then $d\boldsymbol{T} \in \mathbb{R}^{c_o \times c_i}$. And we will have $dt_i = \sum_{j=1}^{c_i} \boldsymbol{T}_{ij}$ for $1 \le i \le c_o$.
- The gradient received by the parameter $\boldsymbol{W}$ is $d\boldsymbol{W} = d\boldsymbol{P} \odot \boldsymbol{M} + d\boldsymbol{P} \odot \boldsymbol{W} \odot H(\boldsymbol{Q}) \odot sgn(\boldsymbol{W})$

Since we add $\ell_2$ regularization in the training process, all the elements in $W$ are distributed within $[-1, 1]$. Meanwhile, almost all the elements in the vector threshold are distributed within $[0, 1]$. The exceptions are the situation as shown in Figure 3(a) and Figure 4(a) where the last layer get no weight pruned (masked). Regarding the process of getting $\boldsymbol{Q}$, all the elements in $\boldsymbol{Q}$ are within $[-1, 1]$. Therefore $H(\boldsymbol{Q})$ is a dense matrix. Then $\boldsymbol{W}$, $H(\boldsymbol{Q})$ and $sgn(\boldsymbol{W})$ are all dense matrices and the pruned (masked) weights can receive the structure gradient $d\boldsymbol{P} \odot \boldsymbol{W} \odot H(\boldsymbol{Q}) \odot sgn(\boldsymbol{W})$

### A.4 Change of remaining ratio in other network architectures

Here we present the change of remaining ratios during dynamic sparse training for the other tested architectures. Since WideResNet models only have one fully connected layer at the end as the output layer, the first five convolutional layers and the last fully connected layer are present. Figure 8 demonstrates the corresponding result for WideResNet-16-8. Figure 9 demonstrates the corresponding result for WideResNet-16-10. And Figure 10 demonstrates the corresponding result for WideResNet-28-8. The similar phenomenon can be observed in all these three network architectures for various $\alpha$.

### A.5 Consistent sparse pattern in other network architectures

Here we present the consistent sparse pattern for the other tested architectures. Figure 11 demonstrates the corresponding result for WideResNet-16-8. And Figure 12 demonstrates the corresponding result for WideResNet-16-10.

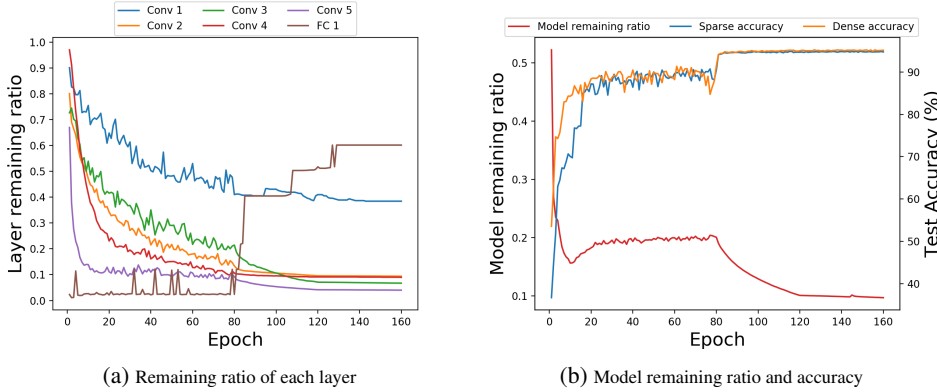

(a) Remaining ratio of each layer  (b) Model remaining ratio and accuracy

Figure 8: WideResNet-16-8 trained by dynamic sparse training with $\alpha = 10^{-5}$

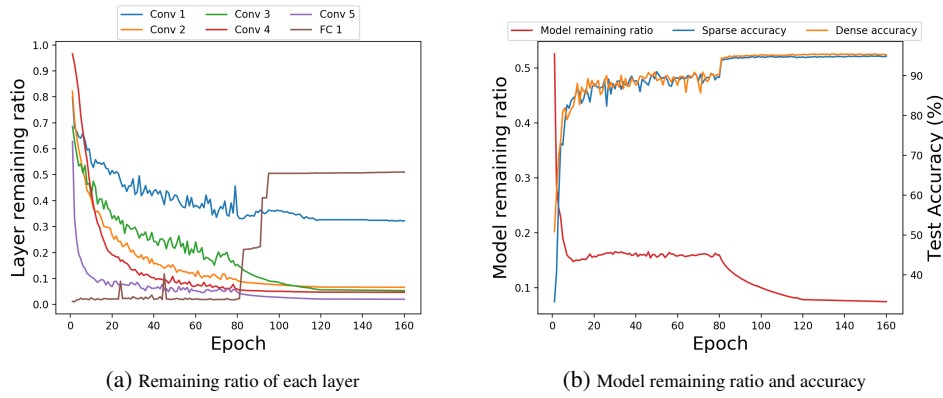

(a) Remaining ratio of each layer

(b) Model remaining ratio and accuracy

Figure 9: WideResNet-16-10 trained by dynamic sparse training with $\alpha = 10^{-5}$

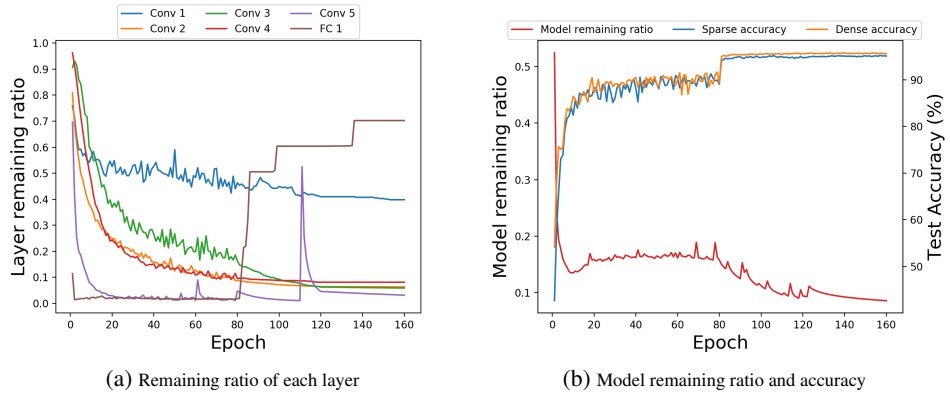

(a) Remaining ratio of each layer

(b) Model remaining ratio and accuracy

Figure 10: WideResNet-28-8 trained by dynamic sparse training with $\alpha = 10^{-5}$

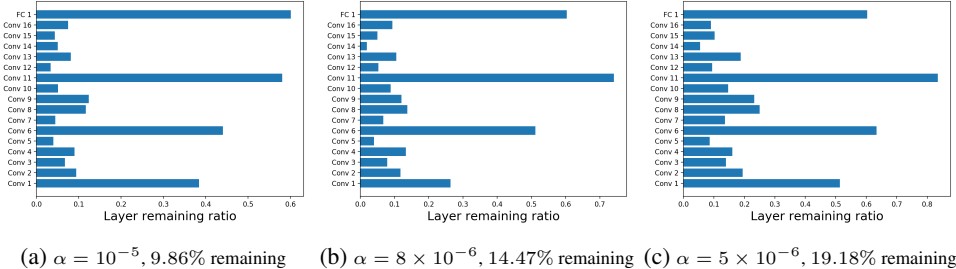

(a) $\alpha = 10^{-5}$, 9.86% remaining  (b) $\alpha = 8 \times 10^{-6}$, 14.47% remaining  (c) $\alpha = 5 \times 10^{-6}$, 19.18% remaining

Figure 11: The sparse pattern and percentage of parameter remaining for different choices of $\alpha$ on WRN-16-8

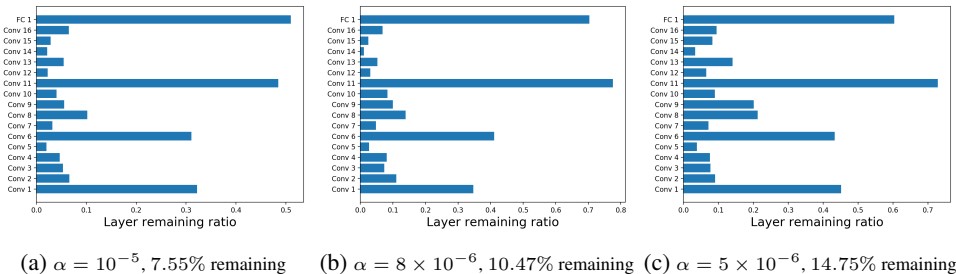

(a) $\alpha = 10^{-5}$, 7.55% remaining  (b) $\alpha = 8 \times 10^{-6}$, 10.47% remaining  (c) $\alpha = 5 \times 10^{-6}$, 14.75% remaining

Figure 12: The sparse pattern and percentage of parameter remaining for different choices of $\alpha$ on WRN-16-10

