# OpenReview forum: "Dynamic Sparse Training: Find Efficient Sparse Network From Scratch With Trainable Masked Layers"
_ICLR.cc/2020/Conference — Accept (Poster)_

### Official Review · AnonReviewer2 · 2019-10-23
**Official Blind Review #2**

**Rating:** 6

**Review:**

## Update after the rebuttal
I appreciate the author's clarification in the rebuttal and the additional result on ImageNet, which addressed some of my concerns.

# Summary
This paper proposes a trainable mask layer in neural networks for compressing neural networks end-to-end. The main idea is to apply a differentiable mask to individual weights such that the mask itself is also trained through backpropagation. They also propose to add a regularization term that encourages weights are masked out as much as possible. The result on MNIST and CIFAR show that their method can achieve the highest (weight) compression rate and the lowest accuracy reduction compared to baselines.

# Originality
- The idea of applying trainable mask to weights and regularizing toward masking out is quite interesting and new to my knowledge.

# Quality
- The performance seems to be good, though it would be more convincing if the paper showed results on larger datasets like ImageNet.

- The analysis is interesting, but I am not fully convinced by the "strong evidence to the efficiency and effectiveness of our algorithm". For example, the final layer's remaining ratio is constantly 1 in Figure 3, while it starts from nearly 0 in Figure 4. The paper also argues that the final layer was not that important in Figure 4 because the lower layers have not learned useful features. This seems not only contradictory to the result of Figure 3 but also inconsistent of the accuracy being quickly increasing up to near 90% while the remaining ratio is nearly 0 in Figure 4.

- If the motivation of the sparse training is to reduce memory consumption AND computation, showing some results on the reduction of the computation cost after sparse training would important to complete the story.

# Clarity
- The description of the main idea is not clear.

- What are "structure gradient" and "performance gradient"? They are not mathematically defined in the paper.

- I do not understand how the proposed method can "recover" from pruned connection, although it seems to be indeed happening in the experiment. The paper claims that the use of long-tailed higher-order estimator H(x) makes it possible to recover. However, H(x) still seems to have flat lines where the derivative is 0. Is H(x) in Equation 3 and Figure 2d are showing "derivative" or step function itself? In any cases, I do not see how the gradient flows once a weight is masked out.

# Significance
- This paper proposes an interesting idea (trainable mask), though I did not fully get how the mask is defined/trained and has a potential to recover after pruning. The analysis of the compression rate throughout training is interesting but does not seem to be fully convincing. It would be stronger if the paper 1) included more results on bigger datasets like ImageNet, 2) described the main idea more clearly, and 3) provided more convincing evidence why the proposed method is effective.

**Experience Assessment:**

I have read many papers in this area.

**Review Assessment: Checking Correctness Of Derivations And Theory:**

I carefully checked the derivations and theory.

**Review Assessment: Checking Correctness Of Experiments:**

I carefully checked the experiments.

**Review Assessment: Thoroughness In Paper Reading:**

I read the paper at least twice and used my best judgement in assessing the paper.

---

> ### Author Response · Authors · 2019-11-06
> **Response to AnonReviewer2  - The concerns about Clarity**
>
> Thank you so much for the detailed reviews and valuable remarks.  I am sorry for the unclarity caused by the lack of information and  improper usage of terms. Here I will use the trainable masked fully connected layer as an example to explain your concerns about Clarity.
> Consider a trainable masked fully connected layer with parameter $W\in R^{m\times n}$ and trainable threshold vector $t\in R^m$. This means that this layer get $n$ input neurons and $m$ output neurons. A neuron-wise threshold $t_i$ is defined for the $i$th output neuron.
>
> 1) How the mask $M\in R^{m\times n} $ is generated and used in the feed forward process
>
> $M_{ij} = S(|W_{ij}|-t_i)$ for $1\leq i \leq m$, $1\leq j \leq n$, where $S(x)$ is the unit step function.
>
> For each connection connects to output neuron $i$, the magnitude of the corresponding weight $W_{ij}$  will be compared with the neuron-wise threshold $t_i$. Instead of directly setting $W_{ij}$ to 0 like traditional pruning algorithms, the value of $W_{ij}$ is preserved in our method. The information about whether to prune this connection is stored in $M_{ij}$, where 0 means pruned (masked) and 1 means unpruned (unmasked). We denote $P = W\odot M$.  Instead of the original parameter $W$,  $P$ will be used in the matrix-vector multiplication.
>
> Meanwhile $Q\in R^{m\times n}$ is just a intermediate variable, where $Q_{ij} = |W_{ij}|-t_i$ for $1\leq i \leq m$, $1\leq j \leq n$
>
> 2) What are "structure gradient" and "performance gradient" mathematically
>
> Refer to Figure 1, in the back-propagation process,  $P$ will receive a gradient and we denote it as $dP$.
> Let's consider the gradients that flow from right to left.
>
> The performance gradient is $dP \odot M$
>
> The gradient received by $M$ is $dP\odot W$
>
> The gradient received by $Q$ is $dP\odot W\odot H(Q)$, where $H(x)$ is the long-tail derivative estimation for $S(x)$ and $H(Q)$ is the result of $H(x)$ applied to $Q$ elementwisely.
>
> The structure gradient is $dP\odot W\odot H(Q)\odot sgn(W)$, where $sgn(W)$ is the result of sign function applied to $W$ elementwisely.
>
> The gradient received by the vector threshold $t$ is $dt\in R^m$. We denote $dT = -dP\odot W\odot H(Q)$, then $dT\in R^{m\times n}$. And we will have $dt_i = \sum_{j=1}^nT_{ij}$ for $1\leq i \leq m$.
>
> 3) How the gradient flow to pruned (masked) weights
>
> The gradient received by the parameter $W$ is $dW = dP\odot M + dP\odot W\odot H(Q)\odot sgn(W)$
>
> Since we add $\ell_2$ regularization in the training process, all the elements in $W$ are distributed within $[-1, 1]$. Meanwhile, almost all the elements in the vector threshold are distributed within $[0,  1]$. The exceptions are the situation as shown in Figure 3(a) and Figure 4(a) where the last layer get no weight pruned (masked). Regarding the process of getting $Q$, all the elements in $Q$ are within $[-1, 1]$. Therefore $H(Q)$ is a dense matrix. Then $W$, $H(Q)$ and $sgn(W)$ are all dense matrices and the pruned (masked) weights can receive the structure gradient $dP\odot W\odot H(Q)\odot sgn(W)$
>
> 4) How the pruned (masked) connection get recovered
>
> The masked weights, the unmasked weights and the vector threshold can all receive gradients and be updated constantly during the training process. A connection with corresponding weight $W_{ij}$ and threshold $t_i$ may be pruned (masked) if $|W_{ij}| < t_i$ at certain time point in the training process. Meanwhile, it can be easily recovered (unmasked) if $|W_{ij}| > t_i$ during the later training process.
>
> 5) Question regarding Equation 3 and Figure 2(d)
>
> The Equation 3 and Figure 2(d) are both present the long-tail derivative estimation. The Figure 2(a) present the unit step function.

---

> ### Author Response · Authors · 2019-11-06
> **Response to AnonReviewer2 - The concerns about Quality**
>
> Thank you for the constructive comments about the quality of our paper.
> Our method achieve state of the art performance compared with other sparse training algorithms due to the fine-grained neuron-wise or filter-wise trainable thresholds together with the proper update of both network parameters and threshold  via back-propagation during the whole Dynamic Sparse Training process. We have published the source code for reproducibility.
>
> To make the illustration more clear, the models that replace the conventional layers with corresponding trainable masked layers and are trained with Dynamic Sparse Training (DST) are referred to as sparse models. For example, LeNet-300-100 with a trainable masked layer and trained with DST are referred to as sparse LeNet-300-100. The dense models train with normal SGD are referred to as dense models. Meanwhile, sparse accuracy indicates the test accuracy after a certain epoch for the sparse models and dense accuracy indicates that for the dense models.
>
> The following are the explanation for your question about the Quality.
> 1) The result on ImageNet
> The result are updated in the revision
>
> 2) The "contradiction" in Figure 3 and Figure 4.
> Thank you for pointing out. it is our negligence to improperly present the experiment results.
> As illustrated in section 4.2, the last layer itself can be regarded as a single layer classiﬁer that takes the features extracted by preceding layers as input. The layer remaining ratio of the last layer can be regarded as an indicator of the eﬀectiveness of the features extracted. If bad features are extracted, the ﬁnal test accuracy will not be high and a relatively large portion of weights in the last layer can be masked. This means that bad features make a large portion of weights in the last layer unimportant. In turn, if a large portion of features extracted is helpful for the classification task, the layer remaining ratio of the last layer as well as the test accuracy should be high.
> Referring to Figure 3(b), the test accuracy of sparse LeNet-300-100 on MNIST reaches over 93% just after the ﬁrst epoch and reaches over 97% after the fourth epoch. Since MNIST is a simple dataset, the model converges very fast. This means that just after the first training epoch, all the features extracted by the preceding layers are somehow helpful for the classification.
> Due to our additional experiment, If a more ﬁne-grained change of layer remaining ratio is present, a similar trend of the remaining ratio of the last layer will be discovered for sparse LeNet-300-100. For example, If the layer remaining ratio after each training step during the first epoch is present, the remaining ratio of the last layer will decrease to a lower value first and then increases up to 1, which is consistent with the trend in Figure 4(a). This phenomenon can be observed with the source code we publish.
>
> 3) More illustration about Figure 4.
> We indeed claim that the parameter importance of the last layer for the sparse VGG-16 model during the first 80 epochs is quite low. However, we also argue that the parameter importance will change dramatically during the training process as the hyperparameters like learning rate change.
> The fact that the accuracy increases up to near 90% before the learning rate decay does not conflict with the fact that most of the features extracted by preceding layers are useless. As present in Figure 4(b), although the remaining ratio of the last layer keeps around 0.05 before the learning rate decay, the test accuracy of the sparse model after each training epoch is better than the test accuracy of the dense model that use all the features extracted.
> We have conducted an additional experiment with the code we publish by training the sparse model without learning rate decay. The accuracy just keeps fluctuating around 85% and the remaining ratio of the last layer also keeps around 0.05. Training more epochs will not increase both the sparse and dense accuracy. If there is no decay of the learning rate, only around 5% of the features extracted are indeed helpful for classification.
> However, if the learning rate is decayed at 80 epoch, the sparse accuracy and the remaining ratio of the last layer increases immediately at the same time as present in Figure 4(b). More useful features are extracted, higher test accuracy will get and the higher remaining ratio will be for the last layer with the decay of learning rate.
> We want to demonstrate the parameter importance may change dramatically and our method can handle this kind of situation properly in Figure 4.
>
>
> 4) Reduction of the computation
> Thank you so much for pointing this out. As you suggest, the quantified result about the reduction of computation and memory should be included. We will add this in the updated version

---

> ### Author Response · Authors · 2019-11-07
> **Response to AnonReviewer2 - Summary of contributions**
>
> Thank you so much for your time and constructive comments.
> It is our negligence that does not clearly present our motivations and contributions of our work in the early manuscript. Here we present the motivations and contributions of our paper.
>
> There are several problems about network pruning that previous methods cannot properly settle:
>
> 1. Expensive pruning and fine-tuning iterations and non-trivial hyperparameters setting
> The pruning and fine-tuning iterations are expensive and need many additional hyperparameters like:
> - How many pruning steps should be adopted
> - How many epochs for the fine-tuning stage after each pruning step
> - Use the same pruning rate or dynamic pruning rate at each step
>
> Our Contribution:
> We propose an end-to-end sparse training process that can get sparse models directly during training without the expensive and non-trivial pruning and fine-tuning iterations.
> There are some similar works but we get better performance than these methods as present in Section 4.1
>
>
> 2. Coarse-grained pruning thresholds
>
> Almost all the previous pruning algorithms adopt a single pruning threshold for each layer or the whole architecture.
>
> Our Contribution:
> In our method, since a threshold vector $t\in R^m$ is used for each layer with parameter $W\in R^{m\times n}$, we have neuron-wise pruning thresholds for fully connected and recurrent layer and filter-wise pruning thresholds for convolutional layer. Usually, there are hundreds of neurons or dozens of filters in a single layer. Thesefore, our method has much more fine-grained pruning thresholds than existing methods.
>
>
> 3. Cannot recover the pruned weights or improper recovery of pruned weights
>
> Most of the current pruning and all the sparse training methods conduct hard pruning with following properties:
> - Pruned weights will be directly set to 0
> - No further update via back-propagation for pruned weights
>
> The importance of weight is not fixed and will change dynamically during the pruning and training process. Previously unimportant weights may tend to be important. So the ability to recover pruned weight is of high significance.
>
> Those sparse learning methods present in Section 4.1 support the recover of pruned weights. However, they all conduct hard  pruning. Directly setting pruned weight to 0 causes the loss of historical parameter importance, which make it hard to determine:
> - When and which pruned weights should be recovered.
> - What value should be assigned to the recovered weights.
> Therefore,  these methods that allow the recovery of pruned weights randomly choose a predefined portion of pruned weights to recover and these recover weights are randomly initialized.
>
> Our contribution:
> - Pruned (masked) weights will not be directly set to 0. Instead, the mask $M$ will store the information about which weights are pruned.
> - Pruned (masked) weights can still be updated via back-propagation.
> - The corresponding pruning thresholds will also be updated via back-propagation
> Therefore, it is the dynamic change of both pruned weights and the corresponding pruning thresholds that determine when and which pruned weights should be recovered. Meanwhile, the recovered weight has the value that it learns via back-propagation instead of a randomly assigned value.
>
>
> 4. How to properly determine the layer-wise pruning rates for complex modern architectures at each pruning step
>
> Most of the existing pruning methods use predefined pruning schedule like like pruning 5% at each step with totally 10 pruning steps for all the network architecture.
> Considering the complexity of modern network architectures, there are dozens of layers that may have various degree of importance and redundancy.  The fixed predefined pruning schedule may not be the optimal choice.
>
> Our contribution:
> The layer-wise pruning rates will be automatically determined by the dynamic change of network parameter $W$ and threshold vector $t$. Meanwhile, the swift adjustment of pruning rates regarding the change of parameter importance present in Section 4.2 and the consistent sparse patterns present in Section 4.3 prove the effectiveness of our method in proper adjustment of layer-wise pruning rates
>
>
> 5. Coarse-grained pruning schedule
> A typical pruning step is conducted after a certain training epoch. Usually, a training epoch will have tens of thousands of training steps, which is the feed-forward and back-propagation pass of a single mini-batch.
>
> Our contribution:
> In our method, since both the network weights and the pruning thresholds will be updated vis back-propagation at each training step. Our method can have continuous fine-grained pruning and recovering at each training step.
> That is our method support step-wise pruning instead of epoch-wise pruning.
>
>
> Thank you again for the detailed review and look forward to your feedback

---

### Official Review · AnonReviewer1 · 2019-10-26
**Official Blind Review #1**

**Rating:** 6

**Review:**

This paper proposes an algorithm for training networks with sparse parameter tensors. This involves achieving sparsity by application of a binary mask, where the mask is determined by current parameter values and a learned threshold. It also involves the addition of a specific regularizer which encourages the thresholds used for the mask to be large. Gradients with respect to both masked-out parameters, and with respect to mask thresholds, are computed using a "long tailed" variant of the straight-through-estimator.

The algorithm proposed in this paper seems sensible, but rather ad hoc. It is not motivated by theory or careful experiments. As such, the value of the paper will be determined largely by the strength of the experimental results. I believe the experimental results to be strong, though I am not familiar enough with this subfield to be confident there are not missing baselines.

There are many minor English language problems (e.g. with articles, prepositions, plural vs. singular forms, and verb tense), though these don't significantly interfere with understanding.

Rounding up to weak accept, though my confidence is low because I am basing this positive assessment on experimental results for tasks on which I am not well calibrated.

more detailed comments:

"using the same training epochs" -> "using the same number of training epochs"
"achieves prior art performance" -> "achieves state of the art performance"
"the inference of deep neural network" -> "inference in deep neural networks"

This paper considered only sparsity of weights -- it might have been nice to also discuss/run experiments exploring sparsity of activations.

eq. 4 -- Can you say more about why this particular form is the right one for the regularizer? It seems rather arbitrary. (it will tend to be dominated by the smallest thresholds, and so would seem to encourage a minimum degree of sparsity in every layer)

I appreciate the analysis in section 4.3.

**Experience Assessment:**

I do not know much about this area.

**Review Assessment: Checking Correctness Of Derivations And Theory:**

I assessed the sensibility of the derivations and theory.

**Review Assessment: Checking Correctness Of Experiments:**

I assessed the sensibility of the experiments.

**Review Assessment: Thoroughness In Paper Reading:**

I read the paper at least twice and used my best judgement in assessing the paper.

---

> ### Author Response · Authors · 2019-11-08
> **Response to AnonReviewer1**
>
> Thank you so much for your time and constructive comments.
>
> It is our negligence that does not present our motivation and contributions of our work clearly in the early manuscript. We are revising it to clearly present our motivations, methods and contributions.
>
> We really appreciate your positive assessment of our experimental results. We are running experiments on the more complex dataset (ImageNet 2012) to make our results more convinced.
>
> Thank you so much for pointing out the language problems in our manuscript. We are polishing the writing continuously and will update the revised manuscript soon.
>
> Following are the responses to the concerns:
>
> 1. The sparsity of activations.
> Usually, only the sparsity of weights is considered for the evaluation of network pruning methods. Your suggestion that we should evaluate the sparsity of activations provides a new point of view. We are conducting related experiments and will add this in the revised manuscript.
>
> 2. The choice of sparse regularizer.
> The sparse regularizer is used to penalize low threshold values to increase the degree of sparsity. So the basic requirement is that the value of the regularizer function $f(x)$ should decrease as $x$ increases.
>
> We actually tested several options like $\exp(-x)$,  $\frac{1}{x}$, and $\log({\frac{1}{x}})$. It seems that other choices except $\exp(-x)$ penalize too much. Therefore the training loss is dominated by the sparse regularizer term $L_s$, which tends to mask out all the weights easily.
> Due to our experiments, $\exp(-x)$ is the best choice among all these options that support wider range choice of $\alpha$ and get higher degree of sparsity. So $\exp(-x)$ is adpoted for the sparse regularizer.  We are still searching for the better sparse regularizer.
>
>
> 3. Sparse regularizer dominated by the smallest thresholds
> Yes, as you point out, the sparse regularizer will be dominated by some small thresholds. Although we discuss the layer-wise sparsity in the paper, the thresholds are actually neuron-wise or filter-wise. Considering a masked fully connected layer with parameter $W\in R^{m\times n}$ and threshold vector $t\in R^m$, this layer will have $m$ output neurons. For each output neuron $i$, our method assigns a neuron-wise threshold $t_i$.
>
> The elements in $t$ are all initialized to $0$, which means that we assume the neurons and weights in this layer have the same importance before the training. And the same penalties for small value are added for all these thresholds.
>
> At the end of the training process, if some thresholds still have small values, it only indicates that these neurons are more important than other neurons so that the weights corresponding to these neurons should have a small degree of sparsity.
> Usually, there are hundreds of neurons in each layer. So the layer-wise sparsity will still be high even if there are few small neuron-wise thresholds.
>
> 4. The analysis in 4.3
> Thank you for your positive feedback about the analysis in section 4.3.
> As we present in section 4.3, our method can generate consistent sparse pattern that indicates the degree of redundancy for each layer.  Besides, our method can distinguish neuron-wise or filter-wise importance with fine-grained neuron-wise and layer-wise thresholds as we present above. Currently, we are not aware of any other method that can have similar effects.

---

### Official Review · AnonReviewer4 · 2019-11-07
**Official Blind Review #4**

**Rating:** 3

**Review:**

This paper presents a novel network pruning algorithm  -- Dynamic Sparse Training. It aims at jointly finding the optimal network parameters and sparse network structure in a unified optimization process with trainable pruning thresholds.  The experiments on MNIST, and cifar-10 show that proposed model can find sparse neural network models, but unfortunately with little performance loss.
The key limitation of the proposed model come from the experiments.
(1) Nowadays, the nature and important question is that, one can not tolerate the degraded performance, even with sparse neural network. Thus, it is important to show that the proposed model can find sparse neural network models, and with increased performance.
(2) Another weakness is that proposed model has to be tested on large scale dataset, e.g. ImageNet-2012.current two datasets are too small to support the conclusive results of this proposed model.
(3) As for the model itself, I donot find very significant novelty. For example, Sec. 3.3 (TRAINABLE MASKED LAYERS) in general is quite following previous works’ designing principle. Thus, the novelty should be summarized, and highlighted in the paper.


**Experience Assessment:**

I have published one or two papers in this area.

**Review Assessment: Checking Correctness Of Derivations And Theory:**

I assessed the sensibility of the derivations and theory.

**Review Assessment: Checking Correctness Of Experiments:**

I assessed the sensibility of the experiments.

**Review Assessment: Thoroughness In Paper Reading:**

I read the paper thoroughly.

---

> ### Author Response · Authors · 2019-11-07
> **Response to AnonReviewer4**
>
> Thank you so much for the constructive comments. We find these comments really helpful. Following are the explanations about the limitations:
>
> 1) The Performance of sparse models.
> Our method indeed gets sparse models with increased performance than dense models.
> As present in section 4.4, our method can get sparse models with better performance when the sparsity of models is less than 90%. It is only when we want to get modes with high sparsity (>90%) that there will be a noticeable performance loss.
> We think this is a common case for network pruning that there will be a loss of performance when over 90% of parameters are removed.
>
> 2) The lack of result on ImageNet
> The results are updated in the revision.
>
> 3) Summary of Novelty
> It is our negligence that does not present the novelty clearly.
> The followings present the novelty of our method:
>
> 1. Directly get sparse models in the training process
>
> The typical pruning process is a three-stage pipeline, i.e., training, pruning and fine-tuning. In our method, no further pruning and fine-tuning are needed. We can directly get sparse models in the training process.
> There exist some similar works but we get better performance compared with the existing methods as present in Section 4.1.
>
>
> 2. Trainable fine-grained pruning thresholds
>
> Most of the previous pruning algorithm adopt a single pruning threshold for each layer or the whole architecture. In our method, since a threshold vector $t\in R^m$ is used for each layer with parameter $W\in R^{m\times n}$, we have neuron-wise pruning thresholds for fully connected and recurrent layer and filter-wise pruning thresholds for convolutional layer.
> Meanwhile, all these fine-grained pruning thresholds can be updated automatically via back-propagation as present in Section 3. We are not aware of any other method that achieves this.
>
>
> 3. The ability to properly recover the previously pruned weights.
>
> Most of the current pruning and all the sparse training methods conduct hard pruning with following properties:
> - Pruned weights will be directly set to 0
> - No further update via back-propagation for pruned weights
>
> Directly setting pruned weight to 0 causes the loss of historical parameter importance, which make it hard to determine:
> - When and which pruned weights should be recovered.
> - What value should we assigned to the recovered weights.
> Therefore,  current sparse training methods that allow the recovery of pruned weights randomly choose a predefined portion of pruned weights to recover and these recover weights are randomly initialized.
>
> Our method has following properties that properly solve these problems:
> - Pruned weights will not be directly set to 0. Instead, the mask $W$ will store the information about which weights are pruned.
> - Pruned weights can still be updated via back-propagation.
> - The corresponding pruning thresholds will also be updated via back-propagation
> Therefore, it is the dynamic change of both pruned weights and the corresponding pruning thresholds that determine when and which pruned weights should be recovered. Meanwhile, the recovered weight has the value that it learns via back-propagation instead of a randomly assigned value.
>
>
> 4. Continuous fine-grained pruning and recovering over the whole training process
>
> A typical pruning process is conducted after a certain training epoch as following:
> - Determine current importance of weight
> - Prune certain percentage of weight.
>
> Usually, a training epoch will have tens of thousands of training steps, which is the feed-forward and back-propagation pass for a single mini-batch.
> In our method, since both the network weights and the pruning thresholds will be updated vis back-propagation at each training step. Our method can have continuous fine-grained pruning and recovering at each training step. We are not aware of any other method that can achieves this.
>
>
> 5. Automatic and dynamic layer-wise pruning rates adjustment over the network
>
> There are two critical problems in network pruning:
> - How many weights should be pruned in each pruning step
> - What is the proper pruning rates for each layer over the network
>
> Usually, the pruning is conduct by some predefined pruning schedule like pruning 5% at each step with totally 10 pruning steps. Meanwhile, it is quite hard to properly determine the pruning rates for each layer. Current methods either use a single global pruning threshold for the whole model or layer-by-layer greedy pruning. We illustrate their limitation on Page 1.
>
> In our method, with the property present above, the portion of weights to be pruned at each step and the proper pruning rates for each layer are automatically determined by the dynamic update of parameter $W$ and threshold $t$. Meanwhile, as present in section 4.2 and 4.3, the swift adjustment of pruning rates and consistent sparse patterns prove the effectiveness of our method in proper adjustment of layer-wise pruning rates.

---

### Author Response · Authors · 2019-11-14
**Summary of Revision**

We appreciate all the detailed reviews and suggestions . Following reviewers' suggestions, we have updated the manuscript and uploaded a revision.  Here we give a summary of the major changes.

 In response to Reviewer 1:

- We present our motivations and contributions more clearly
- We add more experimental results on CIFAR-10 and add the missing result on ImageNet in Section 4.
- We add more analysis about the experimental results on Section 5.1 and 5.2


In response to Reviewer 2:

- Our motivations and contributions are listed more clearly in the introduction part.
- More experimental results on CIFAR-10 and the result on ImageNet are presented in Section 4.1
- We revise the Section 3 to present the main idea more clearly. More details about the feed-forward and back-propagation process are included in Appendix A.2 and A.3. to address the concerns about the  "structure gradient" and "performance gradient".
- We present more details to in Section 5.1 and 5.2 the address the ambiguity of previous edition and  provide more evidences to the effectiveness of our method


In response to Reviewer 4:

- In Section 4.1 and 4.2, we present that  our method is able to get sparse models with increased performance than dense models.
- The results on ImageNet-2012 is present in Section 4.1
- Our motivations and the novelty of our method are summarized and highlighted in the introduction part (Section 1)

---

### Public Comment · ~Aditya_Kusupati1 · 2020-04-17
**Comparison to pruning methods**

Hi,

Great work on trainable masked layers.

My question is about the comparison to DSR and Sparse Momentum methods. Both these techniques are end-to-end sparse training mechanisms, but DST starts off dense and gets dense gradients (always) due to STE and gets to become sparse over time. In this case, won't it be fair to compare against pruning techniques that start out dense and become sparse like global thresholding, iterative magnitude pruning or gradual magnitude pruning?

Let me know if I am missing something.

---

> ### Author Response · Authors · 2020-04-17
> **Reply about the comparison problems**
>
> Hi, Aditya.
>
> Thank you for your attention to our work.
>
> The traditional three-stage pruning algorithms you mentioned usually require many additional fine-tuning epochs compared with normal dense training. One advantage of DST is avoiding the expensive pruning and fine-tuning iterations. DST only requires the same number of training epochs as normal dense training to obtain a sparse model.
>
> To compare fairly with those traditional pruning algorithms, we may need to train the same number of epochs with DST. However, this kind of comparison is not fair either cause increasing the number of training epochs usually won't lead to better performance. The model accuracy tends to saturate after a certain number of training epochs. Those additional fine-tuning epochs are only required for the three-stage pruning algorithms to regain the model performance loss due to pruning operation.
>
> Meanwhile, Sparse Momentum also utilizes the dense gradients to revive the pruned weights.

---

> > ### Public Comment · ~Aditya_Kusupati1 · 2020-04-20
> > **Minor follow-up**
> >
> > I didn't get a notification about the reply, I just saw it now. Thanks for the clarification. A couple of follow-up questions:
> >
> > 1) I agree we need fine-tuning for three state pruning techniques like IMP (Iterative Magnitude Pruning), however, GMP (Gradual Magnitude Pruning) - https://arxiv.org/abs/1902.09574 (a good survey paper) shows that GMP can get good accuracies even for sparse networks in the same number of epochs as normal dense training. Can you please let me know if there is something else here?
> >
> > 2) Yes, Sparse Momentum uses dense gradients (more recent works built on SM also do the same), but it is occasional and periodic which leads to a minimal overhead during training. These methods are not completely sparse-sparse in spirit but are very close to that. I would also like to point you to Discovering Neural Wirings - https://arxiv.org/abs/1906.00586 which uses STE for pruning as well and runs for the same number of epochs as the dense training.
> >
> > To be clear none of them are training the masked layers but rather still rely on sorting. However, their underlying pruning ideology is still the same.
> >
> > Let me know,
> > Aditya

---

> > > ### Author Response · Authors · 2020-04-29
> > > **Reply**
> > >
> > > 1) I think IMP and GMP are basically the same pruning methods. Meanwhile, referring to the original GMP paper (https://openreview.net/pdf?id=Sy1iIDkPM) and the survey paper, it seems that the authors do not clearly indicate that GMP can get good accuracies even for sparse networks in the same number of epochs as normal dense training. Could you please indicate more clearly about this part.
> > >
> > > 2)  Considering the additional overhead for a layer with parameter matrix $W$, Sparse Momentum needs to store and compute the momentum for the parameter of each layer at each training step. Our method only adds a vector threshold for each layer.  Meanwhile, I believe that an implicit mask will be used to prevent updating the pruned weights in practical implementation. So the overall computation overhead of our method is very likely to be less than Spare Momentum.

---

> > > > ### Public Comment · ~Aditya_Kusupati1 · 2020-05-01
> > > > **Reply**
> > > >
> > > > Thanks for the clarification. I agree with 2nd point.
> > > >
> > > > I am not sure about the first point though. What do you mean by "authors do not clearly indicate that GMP can get good accuracies even for sparse networks in the same number of epochs as normal dense training."? I don't understand that statement. The survey paper on GMP (the survey paper does a much better hyperparam sweep for GMP improving results significantly. The original paper's experiments are limited when compared to the survey paper) runs the dense networks and GMP for the same number of epochs. Please check Section 5 of https://arxiv.org/pdf/1902.09574.pdf. They run both dense and sparse training for the same number of epochs.
> > > >
> > > > IMP and GMP are conceptually similar, but their inference costs change vastly due to the allocation of sparsity.
> > > >
> > > > https://github.com/google-research/google-research/tree/master/state_of_sparsity has the models and numbers of all the sparse networks from GMP and they are very comparable to dense training and drops are similar to that of your results.
> > > >
> > > > Any clarification on this would be great. Sorry I missed your session in ICLR.

---

> > > > > ### Author Response · Authors · 2020-05-01
> > > > > **Reply**
> > > > >
> > > > > I have carefully read the survey paper, especially section 5 for several times. The author only mentions that "Each model was trained for 128000 iterations with a batch size of 1024 images". Referring to table 2, there is a sparsity range from 50% to 98%.
> > > > >
> > > > > From my understanding, this means that all the sparse models are trained and pruned with the same configuration. I do not find any clear statement about how the dense baseline is trained and whether the dense baseline is trained with the same training steps. This point is somehow vague. Maybe you can contact the authors of the survey paper for more detailed information.
> > > > >
> > > > > Maybe we need to first figure out whether the GMP could get good accuracies even for sparse networks in the same number of epochs as normal dense training.

---

> > > > > > ### Public Comment · ~Aditya_Kusupati1 · 2020-05-04
> > > > > > **Reply**
> > > > > >
> > > > > > Thanks for getting back to me. I have checked with the authors and it turns out the 1x expts are run for the same epochs as dense baselines and it is the tensorflow google TPU implementation that they used. 1.5x has 50% more epochs than the dense baselines.
> > > > > >
> > > > > > I have used GMP and it does get good accuracies in the same number of epochs as normal dense training. 128000 iterations ~ 90-100 epochs for 1024 batch size which is the typical training time for ResNet50 on ImageNet.

---

### Decision · Program_Chairs · 2019-12-19

**Decision:**

Accept (Poster)

**Comment:**

The paper lies on the borderline. An accept is suggested based on majority reviews and authors' response.